# A high-throughput method for determining the amylose content of rice

**Fei Chen**[ID], **Yunsheng Song, Minghui Dong, Yulin Xie, Zhongying Qiao, Yongliang Zhu, Penghui Cao, Yajie Yu, Caiyong Yuan**[ID]*

Crop Research Department, Institute of Agricultural Science in Taihu Lake District, Suzhou, Jiangsu, China

☯ These authors contributed equally to this work.
* YuanCaiyongN@outlook.com

## Abstract

Rice is one of the world's most abundant food crops. The amylose content is a key factor affecting the production of high-quality rice. The commonly used method for determining the amylose content is the Iodine-Starch colorimetric method. Due to the limitations of traditional iodine colorimetric methods, this experiment provides a simple high-throughput method (SSDM) for determining amylose content. Using rice varieties with different genetic backgrounds as experimental materials, the accuracy and practicality of the new method were explored to improve measurement efficiency and simplify the operation process. The results demonstrated a strong correlation (R = 0.9998) between the absorbance values and amylose content, with a relative deviation of less than 1.2%. It showed good repeatability, with a variation range of 0.43% to 1.63%. The experiment takes a short time and can measure 96 samples at once. Therefore, while ensuring the accuracy of experimental results, the high-throughput measurement method improves the efficiency of measuring amylose content. This has important practical significance for early screening and identification of excellent individuals in rice quality genetic breeding programs.

## Introduction

Rice is one of the most important food crops in the world and the main food source for nearly half of the world's population [1–3]. As the most important producer and consumer of rice, the high requirements for rice quality have been put forward during the production process. The taste and texture of rice are the most important factors affecting rice consumption [2,4,5]. Starch is the main component of rice. According to the different glycosidic bonds connecting D-glucose, starch can be divided into amylose and amylopectin [6]. The texture and sensory quality of the rice are affected by the ratio of amylose to amylopectin. Generally, rice with high amylose content (AC) exhibits a harder texture, with the grains that are fluffy and easily dispersed. On the

**Data availability statement:** All relevant data are within the manuscript and its Supporting Information files.

**Funding:** This study was financially supported by the Research Fund of Suzhou Academy of Agricultural Sciences in the form of a grant received by FC (24024). This study was also financially supported by the Suzhou Science and Technology Project in the form of a grant received by FC (25011). This study was also financially supported by the Key projects of Jiangsu Province's key research and development plan in the form of grants received by ZYQ (BE2021374 and BE2022335-3). This study was also financially supported by the Jiangsu Provincial Fund Project of China in the form of a grant received by CYY (23055). This study was also financially supported by the Suzhou Science and Technology Plan Project in the form of grants received by CYY (SNG202212) and ZYQ (SNG2023004). The funders had no role in study design, data collection and analysis, decision to publish, or preparation of the manuscript.

**Competing interests:** The authors have declared that no competing interesters exist.

**Abbreviations:** AAC, apparent amylose content; BV, the blue value; SDM, national standard; SSDM, simple high-throughput measurement method.

contrary, rice with a good taste is soft and elastic, with shiny grains that are not easy to be separate [7–9]. Therefore, high-quality rice production considers the amylose content of rice as a major indicator of its quality. Nevertheless, current AC determination methods present multiple limitations: they are cumbersome, time-consuming, and labor-intensive in practical applications, resulting in low sample throughput per time unit. Thus, developing rapid and simple amylose detection methods becomes crucial for rice breeding processes [10,11]. Therefore, the establishment of a fast and simple method for detecting amylose is particularly important in the production processes of rice breeding [1–3,11].

Currently, there are many standard methods for determining AC both domestically and internationally, mainly including the following types, iodine affinity method, iodine colorimetric method, high magnification microscopy method, infrared spectroscopy method, and dual wavelength method. Among these, the iodine colorimetric method has been widely used due to its low experimental cost, simple operation, and low experience requirements for operators. Furthermore, different types of national standards differentiate based on whether the test sample undergoes defatting procedures [12–14]. Due to national standards that require degreasing of test sample materials, this process is complex and technically demanding, necessitating large sample volume and testing reagents. Taking all factors into consideration, for processing large quantities of samples, especially in the identification and screening of genetic population samples, the efficiency of experimental operations and the accuracy of detection cannot be guaranteed, which could easily lead to systematic errors [4,9,15].

Given the broad application prospects of the amylose content detection method and its important significance in the evaluation system of grain quality, breeding high-quality varieties, screening as well as identifying excellent germplasm, this study developed a simple high-throughput measurement method (SSDM) [10,14,16]. Integrating the colorimetric reaction with microplate reader technology enhances the accuracy of apparent amylose content (AAC) determination while enabling higher sample throughput. Notably, the pre-treated sample is simple and does not go through the process of starch defatting. The repeatability of the experimental test is good, and the coefficient of variation is small (RSD = 0.43% to 1.63%). The reproducibility of the test sample is high. In addition, it is the small amount of sample used for testing that the gelatinization process of the sample can be completed in a 2.0 mL centrifuge tube. The experimental space is small, the sample gelatinization required less time, and the throughput is high. Each batch can test 96 samples at once, allowing for a large number of test samples. The key innovation of this study lies in the introducing standard samples and absorption peak schemes for amylose content determination, effectively addressing the reproducibility issues caused by reagent or operator variability. To our knowledge, this approach has not been previously reported. Thus, this method holds significant practical value for crop genetic breeding programs. Its high-throughput capability enables efficient mutant screening and early-stage identification of germplasm resources with superior taste quality.

## Materials and methods

### Sample processing and testing principles

After harvesting, the rice was stored in a cool and dry place for a specified period. And then it was milled to obtain polished rice, which was then ground into flour. The samples were selected from near-isogenic lines with different *Wx* alleles [17–19], namely three *indica* rice varieties and three *japonica* rice varieties. Three *indica* rice varieties include Teqing (TQ-*Wx*^a^), 9311-*Wx*^b^, and Yangfunuo (YFN-*wx*). Three varieties of *japonica* rice, 2661-*Wx*^a^, 2661-*Wx*^b^, and Nip-*wx*.

Sample preparation Protocol: **1)** Uniformity of sample particle size. To make the sample particle size, sieve the rice flour with a 100-mesh sieve. **2)** Samples weighing. A standard substance was weighed with a known amylose content of 20 mg (20 mg ± 0.05 mg). The standard samples are respectively numbered from Sample 1 to Sample 4, with the apparent amylose content (AAC) of 1.5%, 10.6%, 16.2% and 26.5%. **3)** Samples dispersion and gelatinization. Then a certain amount of flour was dispersed with absolute ethanol to facilitate gelatinization in 1N sodium hydroxide solution (NaOH) in the later stage; **4)** Samples acidification and acid-iodine compound formation. A certain amount of gelatinized sample solution was taken, and placed in a 1N sodium acetate solution (NaAC) for acidification, and then was added 0.02% iodine solution ($I_2$-KI) reagent to the acidified sample to form a complex between the starch and iodine. **5)** Establishment of iodine absorption spectrum. After measuring the absorbance at a specific wavelength, a linear regression curve was established based on the known AC of the standard samples and their corresponding absorbance values. Based on the linear regression relationship and the absorbance value, calculate AC of the test sample (Figs 1 and 2).

### Reagent preparation

The experimental process requires distilled water, anhydrous ethanol, 1N NaOH, 1N NaAC (pH 4.3), and 0.2% of $I_2$-KI. The preparation of 0.2% iodine potassium iodide reagent requires accurately weighing 20.0 g of KI and 2.0 g of $I_2$, adding an appropriate amount of water, waiting for $I_2$ to fully dissolve, and quantitatively transferring the saturated solution to a 1.0 L volumetric flask. Shake well and make up to volume, then use in the dark. Dispersion preparation includes 200 µL of 1N NaAC solution (pH 4.3), 200 µL of 0.2% $I_2$-KI solution, and 9.0 mL of distilled water. After those are mixed, place them into 10.0mL centrifuge tubes.

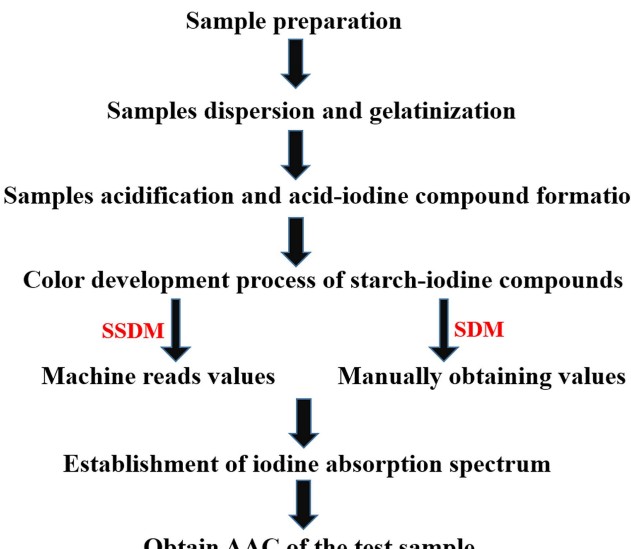

**Fig 1. Method schematic.** SDM, national standard; SSDM, simple high-throughput measurement method.

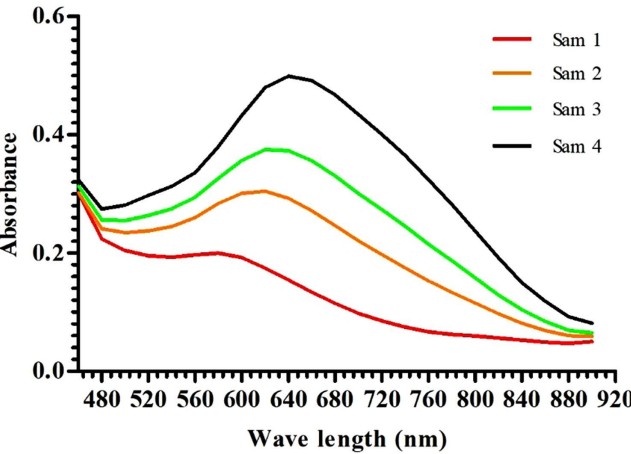

**Fig 2. Iodine absorption spectrum of rice starch.** Sam1~Sam4 represent standard samples 1 to 4, respectively.

## Sample processing and absorption spectroscopy established

The starch -iodine absorption spectroscopy analysis method was referenced from Man et al. and made appropriate modifications [20]. Treat the sample according to the national standard method (SDM), accurately weighed 100 mg ± 0.5 mg flour into a 100 mL conical flask, and carefully added 1 mL of absolute ethanol solution into the sample. Transferred 9 mL of 1 N NaOH into the aforementioned conical flask and shook gently. The mixed sample was then placed in a boiling water bath for 10 minutes, to facilitate sufficient gelatinization. Took out the cooled sample into a 100mL volumetric flask, added water to make up the volume, and shook vigorously. Kept it away from light for 15 minutes. Used a dispersion without samples as a blank control, and used a spectrophotometer (Ultraspec 6300 pro, Amershan Biosciences) to measure the absorbance, with a measurement range of 400–900 nm. The experiment was repeated three times.

The simple high-throughput determination method (SSDM) involved accurately weighing 20 mg ± 0.05 mg of the testing sample into a 2.0 mL centrifuge tube. Added 100 µL anhydrous ethanol to the sample, and then shook the sample evenly. Subsequently, 1.8 mL of 1 N NaOH was transferred to the same 2.0 mL centrifuge tube, and the mixture was vortexed and oscillated to ensure thorough mixing. The testing sample was then placed on a constant temperature shaker set at 60°C to digest at a speed of 200 rpm for 2 hours. During the sample digestion process, the sample dispersion and blank control sample were prepared. After 2 h of digestion, the sample was kept at room temperature. Then, 100 µL of the sample was accurately transferred to a 10 mL centrifuge tube containing the dispersion. After adding the sample, it needed to be shaken up and down. The mixture was left to stand at room temperature for 15 minutes. After standing still, 200 µL of the colorimetric sample was taken and placed into a Costa transparent plate with 96 wells, which was measured on a microplate reader. Each sample was repeated three times, with the dispersion serving as a blank control. The experiment was repeated three times.

## Results and analysis

### Starch-iodine absorption spectrum

The starch-iodine colorimetric method is the classical method for determining the amylose content (AC). The starch-iodine absorption spectrum of the standard samples is shown in Fig 1. Based on the starch-iodine absorption spectrum, we analyzed the maximum absorption wavelength ($\lambda_{Max}$) and absorbance at 620 nm and 680 nm (Fig 2). The blue value (shortened BV) of iodine was used to refer to the absorbance at 680 nm, which represents the affinity between starch

and iodine [21]. Besides amylose, there is also branched starch. However, the structure of amylopectin with varying degrees of polymerization (DP) is relatively complex, consisting of both C-chains with reducing ends and A-chains and B-chains with non-reducing [6,20,22]. The long side-chains with high degrees of polymerization (DP) in branched starch are also compatible with iodine affinity. Therefore, the BV should reflect the apparent amylose content (AAC) of starch [21]. Sample 1 contained glutinous starch. Its starch-iodine absorption spectrum mainly reflected the structural changes of amylopectin. In contrast, the materials used in this experiment (Sample 2, Sample 3, and Sample 4) were selected with different levels of amylose content. From the starch-iodine absorption spectrum, it could be seen that the BV at 680 nm was in the order of Sample 4 > Sample 3 > Sample 2. This could be inferred that the AAC of these samples exhibited the same trend (Sample 4 > Sample 3 > Sample 2). Research has shown that compared to regular starch, branched starch in high amylose has longer side chains. Therefore, the differences between these standard samples may be related to the content of long side chains in amylopectin. Combining experimental data analysis with previous recommendations, 620 nm was selected as the optimal wavelength for starch content determination during measurement.

## Establishment of standard curves for two methods

Four standard samples with amylose content (AC) of 1.5%, 10.6%, 16.2%, and 26.5% were accurately weighed, and their absorbance was measured both using an improved simple high-throughput measurement method (SSDM) and the national standard method (SDM). The SSDM referred to the SDM (GB/T15683-2008/ISO6647–1:2007) and was appropriately optimized based on this. The measured results were plotted as standard curves. The linear correlation coefficient $R^2$ for the SDM was 0.9927, while the linear correlation coefficient for the SSDM was 0.9998. The result analysis indicated that there is a good linear relationship between the absorbance measured by the two methods and the content of amylose (Fig 3). The correlation analysis between the two methods should include significance test results in the figures or tables to confirm that the differences between the results obtained by both methods are not statistically significant. This would provide a stronger basis for subsequent discussions.

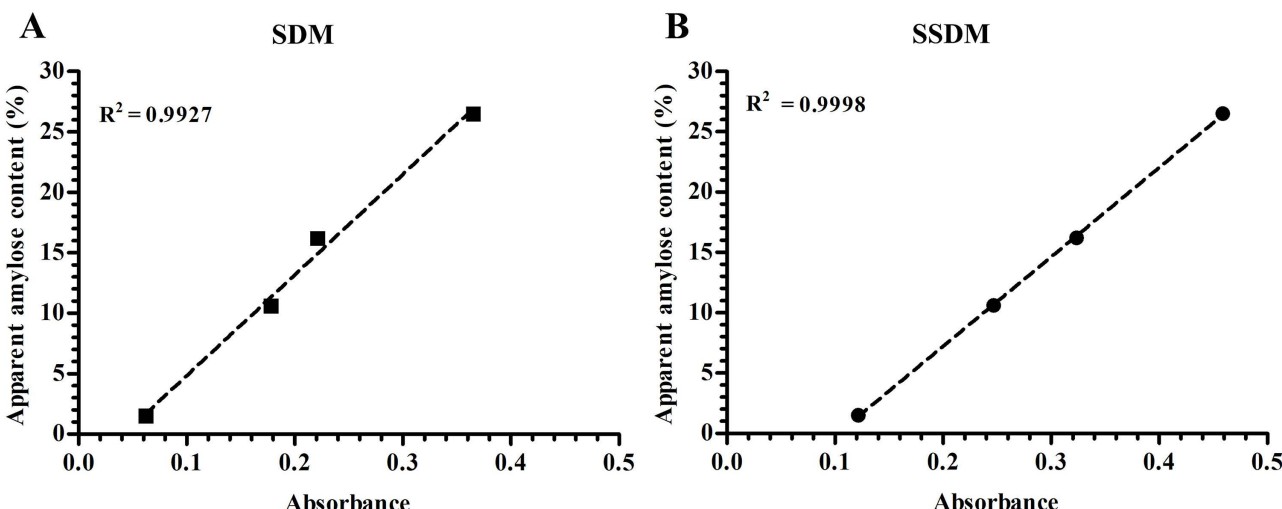

**Fig 3. The standard curves by the methods of SDM ($P$ = 0.0036) and SSDM ($P$ < 0.001).** SDM, national standard; SSDM, simple high-throughput measurement method.

## Correlation analysis between two methods

The amylose content of 6 rice samples was measured using both SDM and SSDM. Each material was subjected to three parallel experiments, and the average of the 3 measured values was the final result (Table 1). Linear regression analysis was performed on the data using SPSS statistical software, with an $R^2$ of being 0.9871, indicating a high correlation between SDM and SSDM for measuring amylose content without a significant difference (Fig 4). Thus, this implies that these findings demonstrate the reliability of SSDM measurement results and confirm that SDM provides methodological and theoretical foundations for SSDM implementation.

## Analysis of influencing factors of SSDM

Many factors affect the determination of starch content, such as the weight of the sample to be tested, the uniformity of the sample, the degree of complete gelatinization of the sample, the concentration of the alkaline solution, and the precision of the measuring sample instrument [18,19,23,24]. To this end, an accuracy analysis of sample weight and digestion time was conducted on the improved SSDM. The analysis results indicate that the overall stability of SSDM tends to be similar (Table 2). When the number of samples was consistent, the accuracy of the SSDM tended to stabilize with the time of starch gelatinization. The accuracy of sample measurements varied complexly within the same digestion time. In addition, the trend in different rice varieties remains consistent across digestion times and sample weights. However, the *indica* rice varieties exhibit a larger coefficient of variation compared to the *japonica* rice varieties, resulting in less stable measurements. This instability may be linked to genetic differences between the two subspecies. Overall, the smaller the

**Table 1. Stability analysis of SSDM under different digestion times.**

| Lines / Time | 0.5 h | 1.0 h | 1.5 h | 2.0 h | 3.0 h | 4.0 h |
|---|---|---|---|---|---|---|
| **TQ-*Wxa*** | 23.17%±0.0059 | 24.91%±0.0109 | 23.74%±0.0033 | 23.69%±0.0019 | 23.77%±0.0038 | 25.56%±0.0054 |
| **2661-*Wxa*** | 21.38%±0.0057 | 23.39%±0.0153 | 23.99%±0.0049 | 20.8%±0.0009 | 22.84%±0.0024 | 23.13%±0.0033 |
| **9311-*Wxb*** | 15.13%±0.0093 | 19.00%±0.0078 | 17.96%±0.0014 | 17.26%±0.0028 | 17.83%±0.0021 | 18.17%±0.0029 |
| **2661-*Wxb*** | 15.29%±0.0066 | 18.79%±0.0065 | 18.91%±0.0090 | 17.02%±0.0017 | 18.32%±0.0023 | 18.66%±0.0021 |
| **YFN-*wx*** | 3.28%±0.0035 | 3.89%±0.0012 | 4.05%±0.0013 | 4.01%±0.0005 | 4.16%±0.0005 | 4.73%±0.0012 |
| **Nip-*wx*** | 2.27%±0.0017 | 2.43%±0.00039 | 3.02%±0.0014 | 2.78%±0.0004 | 2.78%±0.0004 | 2.76%±0.0010 |
| **RSD (%)** | 2.544 - 10.52 | 1.618 - 6.548 | 0.7870 - 4.789 | 0.4289 - 1.631 | 1.037 - 1.579 | 1.100 - 3.473 |

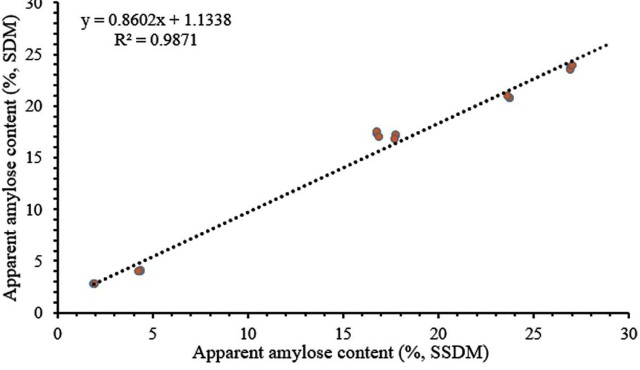

**Fig 4. Correlation analysis between two methods for measuring AAC.** Linear regression analysis of AAC determined SDM and SSDM. SDM, national standard; SSDM, simple high-throughput measurement method ($P=0.0043$).

**Table 2. Precision analysis of the SSDM.**

| Weight \ RSD | RSD$_{0.5h}$ (%) | RSD$_{1.0h}$ (%) | RSD$_{1.5h}$ (%) | RSD$_{2.0h}$ (%) | RSD$_{3.0h}$ (%) | RSD$_{4.0h}$ (%) |
|---|---|---|---|---|---|---|
| 5 mg | 4.28 ± 5.1 Bb | 5.48 ± 2.5 Bb | 3.83 ± 1.9 ABb | 4.73 ± 1.2 ABb | 3.83 ± 1.8 ABab | 3.41 ± 1.5 ABb |
| 10 mg | 3.03 ± 2.6 ABb | 3.59 ± 1.6 ABab | 3.67 ± 1.9 ABab | 2.97 ± 1.1 ABab | 3.14 ± 2.0 ABb | 1.85 ± 0.5 ABab |
| 15 mg | 8.26 ± 2.8 ABb | 4.15 ± 1.0 ABb | 2.89 ± 2.0 ABb | 1.63 ± 1.2 ABb | 2.4 ± 1.3 ABb | 1.74 ± 0.4 ABb |
| 20 mg | 5.63 ± 3.1 ABab | 3.86 ± 1.6 Aa | 2.84 ± 1.6 ABab | 1.11 ± 0.4 ABb | 1.31 ± 0.2 ABab | 2.04 ± 0.8 ABab |

sample amount and the shorter the gelatinization time, the greater the coefficient of variation of the sample determination. Therefore, the smaller the coefficient of variation affecting the determination of amylose when the sample weight is 20.0 mg, the higher the authenticity and reliability of its measurement results.

Analyze the accuracy at different times under different sample weights. The coefficient of variation was represented by RSD. Relative standard deviation, RSD; RSD$_{0.5h}$, RSD$_{1.0h}$, RSD$_{1.5h}$, RSD$_{2.0h}$, RSD$_{3.0h}$, RSD$_{4.0h}$ represent the coefficients of variation corresponding to the samples at 0.5 h, 1.0h, 1.5h, 2.0h, 3.0h, and 4.0h, respectively. SDM, national standard; SSDM, simple high-throughput measurement method. The experiment was repeated three times. Values with different superscript letters differ significantly (ANOVA, $P = 0.05$). Mean ± SD, values following the mean represent the standard deviation (SD).

Sample accuracy analysis. Accurately weigh a sample of 20 mg ± 0.05 mg and measure it at different times. Teqing, abbreviated as TQ- $Wx^a$, Yangfunuo; abbreviated as YFN-wx. Nipponbare-wx was represented by Nip-wx. RSD represents the coefficient of variation. The experiment was repeated three times, mean ± SD.

Sample accuracy analysis. Accurately weigh a sample of 20 mg ± 0.05 mg. Teqing, abbreviated as TQ-$Wx^a$; Yangfunuo, abbreviated as YFN-wx; Nipponbare-wx was represented by Nip-wx. RSD represents the coefficient of variation. R represented repeatability. The experiment was repeated three times, mean ± SD.

## Descriptive analysis of inter-method measurement results

The SDM and the SSDM were used to conduct three repeated measurements on 6 rice samples, and the measurement results were analyzed. According to Table 3, the coefficient of variation of the SSDM ranges from 0.4289% to 1.631%, while the coefficient of variation of the results obtained by the SDM ranges from 0.4248% to 1.1545%. The coefficient of variation in the SSDM was similar to that in the SDM, but there was no significant difference. Moreover, the trend of the determination of amylose content was basically the same. Therefore, it has been proven that the SSDM is reliable and can meet the preliminary determination and screening of large-scale samples.

## Discussion

The endosperm of rice is mainly composed of substances such as starch, protein, oil, vitamins, water, and minerals [3]. Due to the different glycosidic bonds connecting D-glucan, starch is divided into two parts: amylose and amylopectin. The

**Table 3. Comparison and analysis of precision between two methods.**

| Items \ Lines | SDM | | | | | | SSDM | | | | | |
|---|---|---|---|---|---|---|---|---|---|---|---|---|
| | TQ-$Wx^a$ | 2661-$Wx^a$ | 9311-$Wx^b$ | 2661-$Wx^b$ | YFN-wx | Nip-wx | TQ-$Wx^a$ | 2661-$Wx^a$ | 9311-$Wx^b$ | 2661-$Wx^b$ | YFN-wx | Nip-wx |
| AAC (%) | 26.99 ± 0.0 | 23.71 ± 0.0 | 16.81 ± 0.0 | 17.75 ± 0.0 | 4.34 ± 0.0 | 1.94 ± 0.0 | 23.69 ± 0.0 | 20.80 ± 0.0 | 17.26 ± 0.0 | 17.02 ± 0.0 | 4.01 ± 0.0 | 2.78 ± 0.0 |
| RSD (%) | 0.1925 | 0.2192 | 0.3091 | 0.2602 | 1.0634 | 2.3767 | 0.7818 | 0.4289 | 1.6308 | 1.0089 | 1.2808 | 1.4994 |
| r (g/100 g) | 0.5365 | 0.5228 | 0.4881 | 0.4934 | 0.3723 | 0.3170 | 0.5227 | 0.5093 | 0.4906 | 0.4893 | 0.3664 | 0.3405 |
| R (g/100 g) | 1.1545 | 1.0990 | 0.9644 | 0.9845 | 0.5765 | 0.4248 | 1.0987 | 1.0457 | 0.9741 | 0.9689 | 0.5594 | 0.4867 |

variation range of amylose content in rice in nature is relatively narrow, and amylose is one of the important factors determining the appearance quality, cooking and edible quality of rice [6,15,19,25]. Due to its convenient operation, low requirement for experimental operator experience, and low experimental cost, the iodine colorimetric method has become one of the common methods for rice quality identification [2,23,26]. In the experiment, the digestion status of starch affects the accuracy of starch determination and the reproducibility of parallel experiments. The key to measuring the content of amylose is the degree of starch digestion [23]. When the sample dispersion is insufficient, the sample digestion temperature and reaction time are insufficient, which may lead to low starch digestion efficiency, poor experimental reproducibility, and a lack of reference for experimental data, which in turn affects the accuracy of experimental judgment [15]. In the experiment of measuring the amylose content of single or half-grain rice, the phenomenon of insufficient starch gelatinization is particularly prominent. This will limit the application of SSDM in single starch determination. Agasimani et al. crushed and sieved a small amount of single-grain rice samples with embryos removed [12]. The idea of ensuring uniform particle size and consistent sample usage of the sample reduces errors caused by insufficient starch gelatinization. In addition, compared to existing methods for determining amylose content, such as near-infrared spectroscopy and other rapid colorimetric methods, SSDM requires fewer experimental samples. By leveraging the equipment's accuracy and rapid data acquisition capability, SSDM can significantly improve both the throughput and efficiency of sample detection per unit time (Table 4). Therefore, applying this approach to genetic population screening in crop breeding would likely reduce the experimental workload. The correlation coefficient of the SSDM regression curve is 0.9871, indicating a good linear relationship (Figs 2 and 3), thus proving that SSDM can be well applied in starch content determination.

The content of amylose is an important agronomic trait that affects the quality of rice. Many factors affect the content of amylose [1]. Genetic differences are internal factors affecting the content of amylose, and genes related to amylose content mainly include *Wx*, *dull*, and *floury*. Among them, *Wx* is the main gene that controls amylose content, located on the short arm of chromosome 6. The non-glutinous gene *wx* exhibits incomplete dominant inheritance of the glutinous gene *Wx*, and there is a significant dose-response relationship. *Wx* alleles have been reported, including *Wx^a*, *Wx^b*, *Wx^lv*, *Wx^in*, *Wx^op*, *Wx^mp*, and *wx*. These alleles can be distinguished by six functional polymorphic loci, namely Int1–1, Ex2–112, Ex4–53, Ex4–77, Ex6–62, and Ex10–115. In addition, at least 7 different *dull* genes have been discovered, with *du-1* and *du-4*, *du*-2120 and *du*-2035, located on different chromosomes, respectively. Due to the different phenotypes of allelic variations, the amylose content also varies accordingly. To avoid interference with established genetic patterns, this experiment

**Table 4. Comparison of different methods for determining the amylose content.**

| Method Schematic | | SDM | SSDM |
|---|---|---|---|
| **sample preparation** | Sample uniform | 80-100 mesh sieve | 100 mesh sieve |
| | Sample weight | 100±0.5 mg | 20±0.05 mg |
| | Dispersion liquid | 1.0 mL anhydrous ethanol | 0.2 mL anhydrous ethanol |
| **dispersion and gelatinization** | Pasting temperature | 100 °C | 60 °C |
| | Pasting time | 10 mins | 2 h |
| | Gelatinization container | 100 mL conical flask | 2.0 mL centrifuge tube |
| **Color reaction** | Sample dosage | 5.0 mL | 0.1 mL |
| | 1N NaAC | 1.0 mL | 0.2 mL |
| | 0.2% $I_2$-KI | 2.0 mL | 0.2 mL |
| | distilled water | 92.0 mL | 9.5 mL |
| | testing vessels | 1 cm cuvette | 96 cell transparent plate |
| | test equipment | a spectrophotometer | microplate reader |
| | detection wavelength | 720 nm | 620 nm |
| **Test sample quantities** | 2 h per person | 12 samples | 96 samples |

selected 6 representative rice varieties for comparing SDM and SSDM characteristics. There was a linear relationship between starch iodine absorbance and amylose content in the experiment. Therefore, we believe that materials with different genetic backgrounds do not affect the performance of SSDM. Due to the similar weaknesses of SDM and previously reported methods [4,14,15,25,27]. While ensuring the accuracy of the measurement results, we have optimized and improved the SDM to facilitate experimental operations. Compared with previous methods, the SSDM lies in simplifying the operation process and improving the efficiency of sample testing (Table 4). However, since SSDM relies on precision instruments, these limitations may hinder its widespread adoption. Additionally, six materials with known genotypes were selected for this study. The variability of SSDM test results is relatively low (as shown in Table 1, ranging from 0.4289% to 1.631%), which suggests that SSDM may have certain application potential.

The SDM experiment requires a boiling water bath treatment. When there are a large number of samples, the SDM requires too much time [17,27]. However, SSDM is easy to operate, saving the amount of sample and workload investment (Table 4). Simple and effective device modifications not only make the experiment easier to automate operation in a constant temperature oscillating shaker, but also solve the weakness of insufficient dispersion of SDM samples. Compared to the SDM method, SSDM has the following important innovations: 1.) By measuring the absorption wavelength of the standard sample, the most suitable measurement wavelength is selected to ensure the reduction of sample deviation caused by the establishment of the standard curve in the later stage during the operation process; 2.) By utilizing the precision of the microplate reader and the speed of sample detection, the number of sample readings has been increased, while also improving the accuracy of experimental data reading and the convenience of later experimental data recording. Moreover, in rice breeding, SSDM seems unable to identify populations with a low AAC coefficient of variation. These may be due to the presence of a large number of variant individuals with minimal differences in the genetic population. We will seek new solutions in future research to further overcome this challenge. In summary, based on the analysis of existing sample experimental data, the SSDM has the advantages of simple operation, fewer samples, good repeatability, high accuracy and high one-time detection flux, which can be used to identify the offspring of rice and to screen genetic populations in crop breeding. The research on improving crop quality has important practical value.

## Supporting information

**S1 Table. The data for iodine absorption spectrum.**
(XLSX)

**S2 Table. The data for the standard curves.**
(XLSX)

**S3 Table. Measure the AAC of two methods.**
(XLSX)

**S4 Table. Descriptive statistical analysis of SSDM.**
(XLSX)

## Acknowledgments

We greatly appreciate the editors' and referees' great assistance in reviewing the manuscript.

## Author contributions

**Conceptualization:** Caiyong Yuan.

**Data curation:** Fei Chen, Yunsheng Song, Minghui Dong.

**Formal analysis:** Fei Chen.

**Software:** Minghui Dong.

**Visualization:** Fei Chen, Yunsheng Song, Yulin Xie, Zhongying Qiao, Yongliang Zhu, Penghui Cao, Yajie Yu, Caiyong Yuan.

**Writing – original draft:** Fei Chen.

**Writing – review & editing:** Fei Chen, Caiyong Yuan.

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
