## [Decision Letter · Decision Letter 0]

11 Feb 2025

Dear Dr. Yuan,

Thank you for submitting your manuscript to PLOS ONE. After careful consideration, we feel that it has merit but does not fully meet PLOS ONE’s publication criteria as it currently stands. Therefore, we invite you to submit a revised version of the manuscript that addresses the points raised during the review process.

We look forward to receiving your revised manuscript.

Kind regards,

Shailender Kumar Verma, Ph.D.

Academic Editor

PLOS ONE

Journal requirements:   When submitting your revision, we need you to address these additional requirements. 1. Please ensure that your manuscript meets PLOS ONE's style requirements, including those for file naming. The PLOS ONE style templates can be found at https://journals.plos.org/plosone/s/file?id=wjVg/PLOSOne_formatting_sample_main_body.pdf and https://journals.plos.org/plosone/s/file?id=ba62/PLOSOne_formatting_sample_title_authors_affiliations.pdf. 2. Please amend either the title on the online submission form (via Edit Submission) or the title in the manuscript so that they are identical. 3. Please include a caption for figure 2.  4. We note that the grant information you provided in the ‘Funding Information’ and ‘Financial Disclosure’ sections do not match.  When you resubmit, please ensure that you provide the correct grant numbers for the awards you received for your study in the ‘Funding Information’ section. 5. We note that your Data Availability Statement is currently as follows: [All relevant data are within the manuscript and its Supporting Information files.] Please confirm at this time whether or not your submission contains all raw data required to replicate the results of your study. Authors must share the “minimal data set” for their submission. PLOS defines the minimal data set to consist of the data required to replicate all study findings reported in the article, as well as related metadata and methods (https://journals.plos.org/plosone/s/data-availability#loc-minimal-data-set-definition). For example, authors should submit the following data: - The values behind the means, standard deviations and other measures reported;- The values used to build graphs;- The points extracted from images for analysis. Authors do not need to submit their entire data set if only a portion of the data was used in the reported study. If your submission does not contain these data, please either upload them as Supporting Information files or deposit them to a stable, public repository and provide us with the relevant URLs, DOIs, or accession numbers. For a list of recommended repositories, please see https://journals.plos.org/plosone/s/recommended-repositories. If there are ethical or legal restrictions on sharing a de-identified data set, please explain them in detail (e.g., data contain potentially sensitive information, data are owned by a third-party organization, etc.) and who has imposed them (e.g., an ethics committee). Please also provide contact information for a data access committee, ethics committee, or other institutional body to which data requests may be sent. If data are owned by a third party, please indicate how others may request data access.

Reviewers' comments:

Reviewer's Responses to Questions

**Comments to the Author**

1. Is the manuscript technically sound, and do the data support the conclusions?

Reviewer #1: Yes

Reviewer #2: Partly

2. Has the statistical analysis been performed appropriately and rigorously?

Reviewer #1: Yes

Reviewer #2: I Don't Know

3. Have the authors made all data underlying the findings in their manuscript fully available?

Reviewer #1: Yes

Reviewer #2: Yes

4. Is the manuscript presented in an intelligible fashion and written in standard English?

Reviewer #1: No

Reviewer #2: Yes

Reviewer #1: Reviewer’s comment on PONE-D-24-60263

General comment:

The article presents an innovative high throughput method of measuring amylose analysis in rice samples. However, the manuscript was not written in clear, concise and readable English. The advantage/importance of the new method was not clearly presented in the manuscript due to unclear presentation of the materials and method section.

However, I will advise a complete English editing of the manuscript and that the material and method section should be re-written. It was presented as if the authors were writing step by step procedure in a laboratory manual.

Abstract

The abstract should be re-written to follow this pattern:

1. Introductory statement

2. Problem statement

3. Objective of study/statement of objective

4. Materials and method

5. Results and discussion

6. Conclusion

Introduction

There was no introduction section, I only see’ information’ is this the journal’s style?

Line 33-34: clumsy statement, not clear, please revise

Line 47: use ‘defatting instead of ‘degreasing’

Line 57: what is AAC? Write in full

Line 61: ‘it is’ instead of ‘it’s’

Line 63: “due to the fact that’

Materials and Methods

Materials & Method

There must be complete revision of this section to bring out the beauty of the manuscript.

Results, Discussion, Conclusion

Table 3 is not clear

Because of the materials and method section that is muddled up, the beauty, or advantage of the new method is not well elucidated.

The authors highlighted the advantages of the new method, but the methods used to develop the new method is not clearly presented.

Reviewer #2: This article introduces an improved high-throughput method for determining branched starch content based on the national standard. The research findings indicate that the chemical values measured using this method do not exhibit significant differences compared to those obtained through the national standard method.

The article presents valuable findings, but several aspects require improvement. The authors are encouraged to revise the manuscript based on the provided suggestions and refine the language for clarity and coherence.

**Do you want your identity to be public for this peer review?** For information about this choice, including consent withdrawal, please see our Privacy Policy

Reviewer #1: No

Reviewer #2: No

---

## [Author Response · Author response to Decision Letter 1]

19 Mar 2025

Responses to Academic Editors and reviewers' comments

Dear Editors and reviewers,

I sincerely thank you and the two reviewers for their comments and put forward very constructive suggestions and opinions to us. We have carefully reviewed and revised the manuscript. We believe that under the guidance of our beloved editors and review experts, we can give good and appropriate responses to all the comments listed below through our careful modification. The following is a list of major changes based on the opinions of editors and reviewers:

1. According to the comments of editors and reviewers, we highlighted the purpose of the research in the manuscript to facilitate readers' understanding of our work.

2. According to the reviewer's comments, we modified and corrected the content and wording of the manuscript.

3. According to the comments of reviewers, we have made detailed improvements to the materials and method sections of the manuscript.

4. We have carefully revised the manuscript, including correcting typing errors, adding content to explain the data in the chart, and modifying all texts according to the concerns of the reviewers. See below for details.

5. Please include a caption for figure 2.

Response: Thank you to the reviewers for their valuable suggestions. We have revised the section of the manuscript section accordingly, which can be found on page 8, lines 213-214.

6. We note that the grant information you provided in the ‘Funding Information’ and ‘Financial Disclosure’ sections do not match.

Response: Thank you to the reviewers for their valuable suggestions. We have revised the section of the grant numbers for the awards in the ‘Funding Information’ section.

7. We note that your Data Availability Statement is currently as follows: [All relevant data are within the manuscript and its Supporting Information files.]

Response: Thank you to the editors for their valuable suggestions. We have reconfirmed this part of the content and uploaded the original data.

Thanks for your wonderful comments and constructive proposals. The reviewer points out that “The advantage/importance of the new method was not clearly presented in the manuscript due to unclear presentation of the materials and method section.” We have read through the manuscript and made some changes in this, for example, we have revised these sections of the manuscript accordingly, which can be found, and we have deleted a paragraph on page 5. And we also re-organized the paragraphs in page2, lines 33 to 34 as follows: “The texture and sensory quality of the rice are affected by the ratio of amylose to amylopectin. Usually, rice with high amylose content (AC) tends to have a harder texture, with the grains that are fluffy and easily dispersed. On the contrary, rice with a good taste, is soft and elastic, with shiny grains. That are not easy to be separated.”

The purpose of the research introduced in this article is to evaluate three indica rice varieties with varying amylose contents and three japonica rice varieties by comparing the SDM (the national standard method) and SSDM (the improved high-throughput measurement method). The study aims to explore the feasibility of optimizing the SSDM built upon the SDM framework, with a focus on improving measurement accuracy. To enhance experimental efficiency, an innovative enzyme-linked immunosorbent assay (ELISA) was incorporated into the SSDM protocol. This modification not only streamlined the determination process but also reduced operational costs. Additionally, the study analyzed the accuracy and methodological limitations of SSDM during implementation, identifying factors affecting its performance. These innovations and findings address gaps in prior studies on amylose quantification, demonstrating the advantages and application potential of SSDM in cereal science research.

Through our careful efforts to revise and improve, we hope that the revised version can meet the requirements and make the manuscript published in the form of PloS One. Thank you again for your wonderful comments and your valuable time for our manuscript. We sincerely look forward to seeing our working manuscript in your journal.

Reviewer’s comment on PONE-D-24-60263

RESPONSES TO REVIEWER #1

General comment:

This article introduces an improved high-throughput method for determining branched starch content based on the national standard. The research findings indicate that the chemical values measured using this method do not exhibit significant differences compared to those obtained through the national standard method. However, several issues remain in the article, which require clarification from the authors.

Suggested Improvements:

1. Single-Grain Analysis vs. Bulk Sample Analysis

Literature has suggested that conducting rice quality analysis at the single-grain level is more conducive to reducing the breeding cycle of rice. However, the method described in this article processes bulk rice samples to obtain polished rice flour. It is recommended that the authors add an experimental group that processes individual or a few rice grains using the proposed method. The results should then be compared with those obtained using the SDM method to enhance the reliability of the data presented.

References:

L.E. Agelet, C.R. Hurburgh, Limitations and current applications of Near Infrared Spectroscopy for single seed analysis, Talanta, 121 (2014) 288‒299.

Response: This is a valuable suggestion. We had considered such materials in the experimental design. However, since the starch content trait in rice endosperm is classified as an endosperm-specific characteristic following triploid genetic laws, previous studies have discussed this aspect. Given the gene dosage effect observed in single grains or small samples of rice, the sample reproducibility was poor. Therefore, rice noodles (mixed samples) of known varieties were selected for the experiment to avoid single-grain rice as material, thereby minimizing errors caused by gene dosage effects in the endosperm. We plan to further explore analytical methods for single-grain rice in future work.

References:

Zhang AH, Xu CW, Mo HD, Genetic expression of several quality traits in indica-japonica hybrids. Acta Agronomica Sinica 1999, 25, 401-407.

Zhang T, Wang ZR, Mao XC, et al. Research progress of genes affecting rice grain quality. J. Anhui Agric. Sci. 2023, 51, 17-22.

2. Optimization of the SSDM Method

The article claims that the SSDM method is superior to the SDM method. Unfortunately, no schematic or flowchart is provided to illustrate how the method has been optimized. It is recommended that the authors include a logical explanation of the optimization process. Additionally, the description of experimental procedures is overly detailed. Unnecessary steps, such as the sample preparation process in the SDM method, should be removed to better highlight the innovative aspects of the study.

Response: Thank you to the reviewers for their valuable suggestions. We have revised the section of the manuscript section accordingly, which can be found on page 3, lines 89-108.

3. Limited Sample Size

In the "Materials and Methods" section, it is mentioned that only six rice samples were analyzed using both the SDM and SSDM methods, which lacks sufficient representativeness. It is recommended to expand the sample size, including both indica and japonica rice varieties, to improve the generalizability of the SSDM method.

Response: This is a valuable suggestion. We initially considered such materials during experimental design. However, because the Waxy gene regulates starch content in rice endosperm a trait classified as triploid-inherited endosperm characteristics, its mechanism has been extensively documented in prior studies. For instance, Shao et al. reported that many Waxy alleles have been identified in rice, which are Wxa, Wxb, Wxin, Wxop, Wxmq, Wxmw, Wxlv, and wx. The AC of alleles decreases in the following, Wxa (20–25%), Wxb (15–18%) and wx (AC<2%). To avoid interference with established genetic patterns, this experiment selected six representative common rice varieties for comparing SDM and SSDM characteristics. Looking ahead, we will place more emphasis on enhancing the experimental prototype. We will apply this experimental design in future research work.

4. Comparison with Existing Single-Grain Measurement Techniques

Numerous studies have reported well-established single-grain rice measurement methods, including chemical and spectroscopic techniques. The authors should compare their method with these existing techniques to demonstrate its novelty and advantages.

References:

Wu J G, Shi C H. Prediction of grain weight, brown rice weight and amylose content in single rice grains using near-infrared reflectance spectroscopy [J]. Field Crops Research, 2004, 87(1): 13-21.

Yang, J.H., 1992. Study on test methodology of amylose content of single (half) seed of rice. Crop Variety Res. 1, 33–35.

Dai J, Zhang H, Zhu Y, et al. A high–throughput method for determination of apparent amylase content in rice. Acta Agriculturae Zhejiangensis�2014, 26(6): 1421–1424.

Fan S, Xu Z, Cheng W, Wang Q, Yang Y, Guo J, Zhang P, Wu Y. Establishment of Non-Destructive Methods for the Detection of Amylose and Fat Content in Single Rice Kernels Using Near-Infrared Spectroscopy. Agriculture. 2022; 12(8):1258.

Response: This is a very good experimental design idea. We concur with the reviewer that additional studies, data, or further experimental results would be beneficial. Unfortunately, the laboratory does not have the necessary optical measuring instruments. On the other hand, due to SSDM being optimized based on the SDM method.

5. Clarification on the Starch-Iodine Absorption Spectrum (Lines 128–145)

The phrase "starch-iodine" should be corrected to "amylose-iodine."

The article does not specify which wavelength was used for absorbance measurements in the subsequent experiments, which may cause confusion. The authors should clarify this information.

Response: We apologize for causing confusion to the reviewer. In the original manuscript, the discussion regarding ‘Amylose-iodine absorption spectrum’ indicates the selection of experimental testing wavelengths. To offer a more explicit understanding, in the revised manuscript, the corresponding text has been reorganized and modified. It can be located on page 4, lines 142-162.

6. Placement of Results within the Discussion Section (Lines 149–157)

This section presents research results, which should not be included in the discussion. It is suggested that this part be removed or relocated appropriately.

Response: This section has been revised in the article, which this part removed. The removed content can be located on page 5.

7. Significance Testing in Correlation Analysis (Line 169)

The correlation analysis between the two methods should include significance test results in the figures or tables to confirm that the differences between the results obtained by both methods are not statistically significant. This would provide a stronger basis for subsequent discussions.

Response: This section has been revised in the article, which can be located on page 5, lines 174-177. The correlation analysis between the two methods should include significance test results in the figures or tables to confirm that the differences between the results obtained by both methods are not statistically significant. This would provide a stronger basis for subsequent discussions.

8. Precision Analysis of the SSDM Method (Line 193, Table 1)

The authors should discuss differences in measurement precision among various rice varieties (e.g., indica and japonica) under different digestion times and sample amounts.

Response: This section has been revised in the article, which can be located on page 5, lines 200-203. The trend in different rice varieties remains consistent across digestion times and sample weights. However, the indica rice varieties exhibit a larger coefficient of variation compared to the japonica rice varieties, resulting in less stable measurements. This instability may be linked to genetic differences between the two subspecies.

9. Effect of Digestion Time on SSDM Accuracy (Line 197, Table 2)

Standard samples should be used to determine the effect of digestion time on SSDM measurement accuracy, as the amylose content of standard samples is already known.

Response: This is a very good experimental design idea. This section establishes the feasibility of the SSDM method (Figs. 2 and 3), followed by analyzing the influencing factors. The reliability of this method in practical applications is thereby confirmed. Furthermore, the relationship between digestion time and standard sample reliability has been extensively documented in prior research, thus obviating the need for further elaboration.

10. Language and Formatting Issues:

The article contains multiple grammatical errors and redundancies, and it appears that no thorough proofreading has been performed. Examples include:

1) Line 193: "table1" should be written as "Table 1."

Response: I am sorry for my careless mistakes. As suggested by the reviewer, has been corrected in the revised manuscript, as seen in page 7, lines 208.

2) Line 12:

Original: "The simple high-throughput determination method for amylose content is an improved method based on the national standard (GB/T15683-2008/ISO6647-1:2007)."

Suggested Revision: "The improved high-throughput amylose determination method is based on the national standard (GB/T15683-2008/ISO6647-1:2007)."

Response: This is a very good idea. This section has been revised in the article. As suggested by the reviewer, has been corrected in the revised manuscript, as seen in page 2, lines 4-5.

3) Line 20:

Original: "There is no significant difference between the amylose content measured using this method and the values obtained through the national standard method."

Suggested Revision: "The amylose content measured by this method shows no significant difference from that obtained through the national standard method."

There are numerous other instances of similar issues that require careful revision.

Response: Thank you very much for the suggestions from the reviewing experts. This section has been revised in the article, as seen in page 2, lines 42-43.

11. Some of the references are outdated and should be replaced with more recent literature.

Response: We totally agree with the reviewer. In the revised manuscript, they have been added as the references as seen in pages 10-12, lines 293-355, meanwhile the other related references have also been referred.

Conclusion:

The article presents valuable findings, but several aspects require improvement. The authors are encouraged to revise the manuscript based on the provided suggestions and refine the language for clarity and coherence.

RESPONSES TO REVIEWER #2

The comments are as follows:

The article presents an innovative high throughput method of measuring amylose analysis in rice samples. However, the manuscript was not written in clear, concise and readable English. The advantage/importance of the new method was not clearly presented in the manuscript due to unclear presentation of the materials and method section.

However, I will advise a complete English editing of the manuscript and that the material and method section should be re-written. It was presented as if the authors were writing step by step procedure in a laboratory manual.

1. Abstract

The abstract should be re-written to follow this pattern:

1. Introductory statement

2. Problem statement

3. Objective of study/statement of objective

4. Materials and method

5. Results and discussion

6. Conclusion

Response: Thank you to the reviewers for their valuable suggestions. We have revised the 'Abstract' section of the manuscript accordingly, which can be found in page 1, lines 29-45.

2. Introduction

There was no introduction section, I only see’ information’ is this the journal’s style?

Response: We sincerely apologize for the oversight, as w

---

## [Decision Letter · Decision Letter 1]

27 May 2025

Dear Dr. Yuan,

Thank you for submitting your manuscript to PLOS ONE. After careful consideration, we feel that it has merit but does not fully meet PLOS ONE’s publication criteria as it currently stands. Therefore, we invite you to submit a revised version of the manuscript that addresses the points raised during the review process.

We look forward to receiving your revised manuscript.

Kind regards,

Karthikeyan Venkatachalam, Ph.D.

Academic Editor

PLOS ONE

Additional Editor Comments :

Dear Authors

Thank you for your revised version of the manuscript. Although you have addressed all the given comments from the reviewers extensively, and revised it accordingly in the manuscript, and however, I remain feel that your manuscript still requires improvements based on the following comments. Please revise it accordingly.

1. Please address the following comments in order to make the final decision for your manuscript.

2. The study addresses a relevant challenge in rice quality evaluation by proposing an improved high-throughput method for determining amylose content (SSDM), building upon an existing standard.

3. The novelty of the SSDM method is moderately articulated. While the authors claim operational efficiency and reduced labor, the mechanistic or methodological innovation beyond equipment change and protocol condensation is insufficiently developed. The authors should clarify what is scientifically novel beyond throughput or procedural efficiency.

4. The experimental design is limited by a very small sample size (six rice varieties), reducing the generalizability and statistical power of the conclusions. The authors must justify the small sample size or expand it. Include rationale for the selection of the specific varieties.

5. The comparison to the standard method lacks critical methodological controls, particularly regarding standardization of sample preparation and instrument calibration across both methods. Provide a detailed explanation of how comparability was ensured.

6. Statistical rigor is compromised by the absence of clear reporting of p-values or confidence intervals in correlation and regression analyses. Include appropriate statistical tests (e.g., t-test or ANOVA) to support claims of no significant difference.

7. The abstract is verbose and poorly structured, with overlapping and redundant sentences. Revise the abstract to follow a standard format (Background, Objective, Methods, Results, Conclusion).

8. The language throughout the manuscript suffers from frequent grammatical errors, awkward phrasing, and unclear sentence structures. Conduct professional English editing and sentence restructuring.

9. Key experimental variables (e.g., digestion temperature, NaOH concentration) are introduced but not thoroughly tested. Include justification for selected values or conduct sensitivity tests.

10. Table formatting, especially in Tables 1–3, is suboptimal. Improve clarity, define abbreviations, and ensure all values are appropriately labeled with units.

11. The explanation of how SSDM integrates with ELISA technology is vague and potentially misleading. Clarify if ELISA is truly used. If not, remove the term to avoid confusion.

12. SSDM’s reliance on a 96-well plate format and spectrophotometric detection is not novel. Authors should specify what distinguishes SSDM from existing high-throughput assays.

13. Figure legends are inadequately described, and essential figures (e.g., method schematic) are missing. Add a schematic comparing SDM and SSDM and revise all figure captions to be self-explanatory.

14. The justification for excluding single-grain analysis is weak. Provide stronger evidence or literature to support this decision, or present it as a limitation.

15. References are outdated in several places. Update references with recent publications from the past 5 years.

16. The discussion section repeats results without deeper interpretation. Expand discussion to include scientific implications, breeding relevance, and future directions.

17. The manuscript does not adequately address performance on unknown or genetically diverse samples. Clarify whether SSDM is applicable to diverse rice germplasm. Acknowledge as a limitation if untested.

18. The conclusion overstates the practical implications of SSDM. Moderate the conclusion and align it with the actual findings and scale of validation.

19. Although the SSDM exhibits low variability in repeated measures, the magnitude of improvement over SDM is marginal. Quantify the gains in throughput, time, or cost explicitly.

20. No limitations section is included. Add a section discussing limitations (e.g., dependency on known genotypes, lab setup requirements, variability).

21. The authors' response to prior reviewer critiques is extensive but lacks depth in several places. Responses should go beyond wording edits and address core methodological concerns.

22. Claims about SSDM's high throughput would benefit from benchmarking against commercial kits or rapid analytical platforms. Compare SSDM with commercial or literature methods in terms of speed, sensitivity, and cost.

23. SSDM’s reliance on microplate reader and shaker may limit its use in field labs. Acknowledge instrumentation limitations and discuss potential adaptations.

24. The acronym “SSDM” includes ELISA, which is not implemented in the actual method. Correct acronym expansion or remove misleading reference to ELISA.

25. The SSDM method appears suitable for preliminary screening but lacks validation across diverse use cases. Clearly define current validation scope and outline required future studies for broader adoption.

26. In Table 1, the authors present RSD ranges in descending order (e.g., “2.5410.53”). This is unconventional; standard scientific reporting typically presents RSD ranges in ascending order, please correct it.

Reviewers' comments:

Reviewer's Responses to Questions

**Comments to the Author**

Reviewer #1: All comments have been addressed

2. Is the manuscript technically sound, and do the data support the conclusions?

Reviewer #1: Yes

3. Has the statistical analysis been performed appropriately and rigorously?

Reviewer #1: Yes

4. Have the authors made all data underlying the findings in their manuscript fully available?

Reviewer #1: Yes

5. Is the manuscript presented in an intelligible fashion and written in standard English?

Reviewer #1: Yes

Reviewer #1: I have gone through the revised version of this manuscript The authors have carried out the suggested revisions. The manuscript can be accepted now

**Do you want your identity to be public for this peer review?** For information about this choice, including consent withdrawal, please see our Privacy Policy

Reviewer #1: No

---

## [Author Response · Author response to Decision Letter 2]

13 Aug 2025

Please refer to the reply attachment titled 'Response to Reviewers 20250723'.

---

## [Editor Report · Decision Letter 2]

25 Aug 2025

Dear Dr. Yuan,

We look forward to receiving your revised manuscript.

Kind regards,

Karthikeyan Venkatachalam, Ph.D.

Academic Editor

PLOS ONE

**Journal Requirements:**

**Additional Editor Comments:**

Dear Authors,

Thank you for submitting the revised version of your manuscript. Before the manuscript can be considered for acceptance, the following mandatory revisions must be made:

1. Clarify and strengthen the novelty of SSDM compared to existing amylose determination methods (e.g., NIR, other rapid colorimetric adaptations). Explicitly state the research gap that SSDM addresses.

2. Revise the methods section for reproducibility. Remove all formatting artifacts and duplicated text, and clearly state whether replicates are biological or technical. Ensure the protocol is transparent enough for replication.

3. Provide proper statistical analysis beyond RSD values. Apply appropriate tests (e.g., ANOVA, t-tests) to support claims of superiority and report corresponding significance levels.

4. Expand the limitations section to acknowledge the small validation set (six rice varieties), the dependency on specialized instruments such as a microplate reader, and the lack of validation across broader varietal or environmental conditions.

5. Moderate the conclusions. Restrict claims to what is directly supported by the dataset and avoid overstating applicability or adoption potential.

6. Correct all terminology inconsistencies (e.g., misuse of ELISA terminology, inconsistent references to “starch-iodine” vs “amylose-iodine”) and ensure uniform usage throughout.

---

## [Author Response · Author response to Decision Letter 3]

14 Oct 2025

Please refer to the reply attachment titled 'Responses to PloS One academic reviews1 1012'.

---

## [Editor Report · Decision Letter 3]

28 Oct 2025

A high-throughput method for determining the amylose content of rice

PONE-D-24-60263R3

Dear Dr. Yuan,

We’re pleased to inform you that your manuscript has been judged scientifically suitable for publication and will be formally accepted for publication once it meets all outstanding technical requirements.

Kind regards,

Karthikeyan Venkatachalam, Ph.D.

Academic Editor

PLOS ONE
---

## [Editor Report · Acceptance letter]

PONE-D-24-60263R3

PLOS ONE

Dear Dr. Yuan,

I'm pleased to inform you that your manuscript has been deemed suitable for publication in PLOS ONE. Congratulations! Your manuscript is now being handed over to our production team.

Kind regards,

on behalf of

Dr. Karthikeyan Venkatachalam

Academic Editor

PLOS ONE